# Habitats and Biotopes in the German Baltic Sea

**DOI:** 10.3390/biology13010006

**Published:** 2023-12-21

**Authors:** Denise Marx, Agata Feldens, Svenja Papenmeier, Peter Feldens, Alexander Darr, Michael L. Zettler, Kathrin Heinicke

**Affiliations:** 1Leibniz Institute for Baltic Sea Research Warnemünde, Seestrasse 15, D-18119 Rostock, Germany; a.feldens@subsea-europe.com (A.F.); svenja.papenmeier@io-warnemuende.de (S.P.); peter.feldens@io-warnemuende.de (P.F.); alexanderdarr@web.de (A.D.); michael.zettler@io-warnemuende.de (M.L.Z.); 2German Federal Agency for Nature Conservation, Isle of Vilm/Rügen Office, D-18581 Putbus, Germany; kathrin.heinicke@bfn.de

**Keywords:** habitats, biotopes, mapping, MSFD, broad habitat types—BHTs, other habitat types—OHTs, HELCOM Underwater biotope and habitat classification system—HUB, predictive biotope modelling, Baltic Sea

## Abstract

**Simple Summary:**

This study provides full-coverage maps of the habitats and biotopes in the German Baltic Sea at an unprecedented level of resolution. We combined geological and biological surveys to map the seabed and collected extensive data to classify different habitats and their inhabitants. Using newly established national guidelines and modelling, we produced highly accurate maps. These maps are of practical use in meeting national and regional reporting requirements, facilitating management decisions, supporting marine spatial planning, and answering research questions.

**Abstract:**

To maintain or enhance biodiversity and sea floor integrity, mapping benthic habitats is a mandatory requirement in compliance with the Marine Strategy Framework Directive (MSFD). The EU Commission Decision distinguishes between Broad Habitat Types (BHTs) and Other Habitat Types (OHTs). At the regional level, biotopes in the Baltic Sea region are classified according to the HELCOM underwater biotope and habitat classification (HUB). In this study, the habitats and their benthic communities were mapped for the entire German Baltic Sea at a high spatial resolution of 1 km. In two nature conservation areas of the Exclusive Economic Zone (EEZ) as well as selected focus areas in the coastal waters, the resolution we provide is even more detailed at 50 × 50 m. Hydroacoustic data recording and benthological surveys (using bottom grabs, underwater towing camera technology, and diver sampling) helped identify biotopes in high resolution. Based on these data, together with additional data acquired since 2010 (a total of over 7000 stations and transect sections), we were able to spatially delineate benthic biotopes and their communities via predictive habitat modelling. The results are provided as full-coverage maps each for BHT, OHT, and HUB (9 classes of BHTs, 5 classes of OHTs, and 84 classes of HUB) with a level of spatial detail that does not yet exist for the Baltic Sea, and they form an essential basis for future monitoring, status assessments, and protection and management measures.

## 1. Introduction

A good environmental status in our oceans is more important than ever. The Baltic Sea is particularly vulnerable to anthropogenic pressures due to its unique and fragile ecosystem (involving marine and limnic influences, a shallow depth, and limited water exchange through the shallow Strait system) [1,2]. Over the past century, the Baltic Sea has experienced human-induced regime shifts towards a eutrophic state with altered species composition [3], affecting entire food webs [4]. To overcome transnational challenges, regional regulatory frameworks were created to protect marine ecosystems. The Marine Strategy Framework Directive (MSFD, 2008/56/EC [5]) was initiated at the EU level to protect, conserve, and, where feasible, restore the marine environment. Among other things, the Directive requires EU member states to assess and monitor the current ecological status of their marine waters, with the aim to implement measures to reduce pressures as well as monitor their effectiveness. Benthic habitats and their specific benthic communities (together considered as biotopes) are of ecological importance as integral parts of the food web, providing breeding, nursery, and feeding grounds for benthic and pelagic species, as well as migratory birds, and may even represent a biodiversity hotspot [6]. Consequently, the Directive considers the seafloor and its inhabitants under Descriptor 1 (“Benthic Habitats”) and Descriptor 6 (“Seafloor Integrity”) with the view that benthic ecosystems and physical disturbance as well as loss of the seabed should be avoided, and that they should not be adversely affected (MSFD Annex I). To implement the requirements of the MSFD, but also other EU legislation and regional programmes, e.g., the Habitats Directive (HD, Article 11, 92/43/EEC [7]) and the Baltic Sea Action Plan set by HELCOM (Baltic Marine Environment Protection Commission, or shortly, Helsinki Commission), reliable full coverage maps of the distribution and extent of benthic habitats as well as their changes over time are essential. The respective regulations are based on different habitat and biotope classification systems which have been adapted to each other over time. For the Habitats Directive, delimitation rules for habitat indication had to be created [8], while the MSFD is based on the European Nature Information System (EUNIS), and HELCOM developed its own system (HUB—HELCOM Underwater Biotope and Habitat Classification [9]).

The assessment within the framework of the MSFD must be carried out separately for benthic broad habitat types (BHTs) and other habitat types (OHTs) [10]. Examples of BHTs, which are specified according to the EU Commission Decision (2017/848/EU [11]), are “Infralittoral/circalittoral rock and biogenic reef”, “Infralittoral/circalittoral mixed sediment”, “Infralittoral/circalittoral coarse sediment”, “Infralittoral/circalittoral sand”, and “Infralittoral/circalittoral mud”. The delimitation of these habitats is based on the definition of the European Nature Information System, EUNIS, and corresponds to EUNIS level 2 (ending at level 3 with regard to the Baltic Sea). EUNIS covers, in a hierarchical way, marine benthic habitats with their sedimentological and biological components. Infralittoral refers to the light-flooded zone that allows for the growth of vascular plants and green algae [12]. In contrast, the circalittoral is not sufficiently flooded with light. EUNIS also distinguishes between the offshore circalittoral and the (nearshore) circalittoral. However, the offshore circalittoral is not relevant in Germany.

Additional benthic habitat types (so-called other habitat types, OHTs) can be selected by the respective EU member state to be assessed separately from BHTs, e.g., due to their ecological relevance as protected biotopes [10]. Such OHTs for German marine waters include biotope types according to §30 of the German Federal Nature Conservation Act (BNatSchG), according to the European Habitats Directive (92/43/EEC), as well as the Baltic Sea-wide HELCOM Red List types [13]. Relevant OHTs of the German Baltic Sea are “species-rich areas of gravel, coarse-sand and shell-gravel areas”, “seagrass meadows and other marine macrophyte populations”, “reefs”, “sandbanks which are slightly covered by sea water all the time”, and “Baltic aphotic muddy sediment dominated by ocean quahog (*Arctica islandica*)”.

The latter biotope originates from the HELCOM HUB system. It is a hierarchical system, in which the Baltic Sea marine region, the light availability, the structuring habitat, and the associated dominant benthic community are entered and indicated in a uniform code. The definition of the HUB system is largely compatible with EUNIS. In contrast to the BHTs, which are equivalent to EUNIS level 2 (the substrate level), the HUB biotope types in this study are specified down to level 6 (dominating taxa), i.e., the lowest level possible. EUNIS level 2 (or BHT) is consistent with HUB level 3.

Benthic habitat types according to the Habitats Directive [14] and EUNIS [15] as well as benthic biotope types [16,17] or both [18] were previously mapped for selected parts of the Baltic Sea. Habitat types as full-coverage maps were provided by the EMODnet Seabed Habitats project for the entire Baltic Sea on a large scale [19]. A comparable map with HUB biotope types for the German Baltic Sea has previously only been presented by Schiele et al. [20] and modified by Zettler and Darr [21]. However, these maps were based on a limited dataset and took little account of the epibenthic colonisation of the hard substrate. Since the compilation of the map by Schiele et al. in 2015 [20], guidelines were developed on how to geologically map the seafloor in German marine areas in a standardised way, how to compile sediment and boulder distribution maps, and how to delineate geogenic reefs on a large scale, respectively [8,22]. Selected areas in this present study were comprehensively hydroacoustically mapped at a high resolution, and sediment types were delineated using these mapping instructions from the BSH (Federal Maritime and Hydrographic Agency) and the BLANO technical expert group, HyMo (technical expert group “Hydrography, Hydrology and Morphology” of the Federal Government/Federal States Committee on the North Sea and Baltic Sea). Newly developed AI-supported methods for the semi-automatic detection of boulders supported the updated mapping [23,24,25]. Various sources of information were used in order to create maps that are as coherent as possible and to be able to indicate hard bottom communities in particular. Thus, the BHT, OHT, and HUB maps reach an unprecedented level of detail, combining various classification systems with updated data, which can now be reproduced in a standardised way.

This study provides the basis for the national assessment under MSFD Descriptor 6, taking into account the requirements of various regulations, and it is thus vital for further management decisions and the adaptation of monitoring programmes.

## 2. Materials and Methods

The study area covers the entire German part of the Baltic Sea. However, the map showing HUB biotopes could not be produced for the inner coastal waters of Mecklenburg-Western Pomerania due to a lack of data. The created maps were grid-based with a spatial resolution of at least 1 × 1 km. In areas where seabed sedimentology was fully mapped using hydroacoustic techniques (following section; Figure 1), namely the “Outer Wismar Bay”, the “Darss Sill”, and the “Plantagenet Ground” in the coastal waters of Mecklenburg-Western Pomerania, as well as the nature conservation areas “Fehmarn Belt” and “Kadet Trench” in the EEZ, a resolution of 50 × 50 m was applied. These high-resolution areas that were mapped with hydroacoustic recordings are referred to as “detail areas” in the following sections. For more information on each detail area (the sediment and boulder distribution maps as well as BHT, OHT, and HUB maps), see [25,26,27,28,29].

### 2.1. Geological Mapping

The sedimentology of the seafloor was mapped by hydroacoustic methods in three areas in the coastal waters of Mecklenburg-Western Pomerania and two nature conservation areas in the EEZ [25,26]. Side-scan sonars (including Marine Klein 4000, Edgetech 4200, Edgetech 4300 MPX, Starfish 450F, Edgetech 4200 HF, and R2Sonic2024) with different frequencies (100–600 kHz) were used for this purpose. External data were acquired (Vermessungsbüro Weigt, BSH, Christian-Albrechts-University Kiel) in order to apply the time-consuming measurements only in areas where data with the required resolution and quality were not yet available. Sediment samples and videos were used to verify the hydroacoustically recorded data within the detail areas (ground truthing, as shown in Section 2.5 and further described together with the benthos sampling in Section 2.4). Data processing was carried out with SonarWiz (Chesapeake Technology Inc., Los Altos, CA, USA) software, which creates backscatter mosaics of the seafloor surface. After setting the bottom tracks, correcting for slant range distortion, and setting the layback, empirical gain normalisation was applied, and the backscatter mosaics were imported into ArcMap 10.7.

Sediment analyses from ground truthing samples were performed optically using a Mastersizer 3000 (Malvern Panalytical Ltd., Malvern, UK) as well as by dry and wet sieving due to the heterogeneity of the samples (silt, sand, and coarse sediment). The Mastersizer was used to measure samples up to a maximum grain size of 3.0 mm. Samples with coarser components were sieved. Seemingly fine-grained samples were treated with HCl and H_2_O_2_ before measurement to remove carbonate and organic components, because these compounds impact grain size distributions. For the evaluation of the hydroacoustic data, the results of the sediment analyses were evaluated according to Blott and Pye [30] and fitted to the Folk triangle [31] while considering the BSH hydroacoustic mapping instructions [22].

With the help of the hydroacoustic data, video recordings, sediment samples, and the sediment map according to Tauber [32], sediment distribution maps were created within the detail areas according to a national guideline [22] with the following standards: Sedimentological classification of the areas delineated on the backscatter mosaic was performed for the sediment types at three different levels where possible. Level A includes fine sediments, sands, mixed sediments, coarse sediments (simplified classification according to Folk [31]), and both residual sediments and peat. The term “residual sediment” (lag sediment) is not a clearly defined concept in marine geosciences, but it is nevertheless frequently used for a higher-level description of certain sediment types. Residual sediments cannot be identified by an exact granulometric measurement. Rather, residual sediments describe the remaining part of sediments that have been reworked by natural dynamic processes. Such reworking processes usually result in a granulometric separation/sorting of the sediment components, whereby the less mobile components like gravel, small boulders, or marl remain in the area of the original sediment for longer [33,34,35]. This distinguishes them from the categories of mixed sediments and coarse sediments defined by Folk [31], which contain only mud, sand, and gravel, but not fractions beyond that. In Level B, the clastic sediment types from Level A are further subdivided according to Folk [31]. Since there is no subdivision of sands in the Folk triangle, Level C was introduced, in which sediments designated as sands in Levels A and B were further subdivided according to Figge [36] (for more information, see [26]; Table 1).

The distribution of boulders is displayed in a separate map. The number of boulders in the nature conservation areas in the EEZ (“Fehmarn Belt” and “Kadet Trench”) was estimated manually for each 50 × 50 m grid cell by dividing them via subitising (recording the number of boulders at first sight without counting) into three classes, according to the guideline for the large-scale delineation of geogenic reefs in the German Baltic Sea [8]: cells without boulders (Class 1), cells with 1–5 boulders (Class 2), and cells with more than 5 boulders (Class 3). Boulders in the detail areas of the coastal waters in Mecklenburg-Western Pomerania (“Outer Wismar Bay”, “Darss Sill”, and “Plantagenet Ground”) were detected semi-automatically using the methods reported in the study by Feldens et al. [23,24]. A neural network detected individual boulders in the backscatter mosaics [25]. Where possible, mosaics acquired at a frequency between 300 and 500 kHz were chosen as a baseline, as these show better individual object resolution [37]. The results were screened for false detections (mainly caused by water column stratification artefacts in the data) and then classified into the same three classes as described above. A boulder distribution map was used to place a grid over the areas and indicate these three categories per 50 × 50 m cell.

The sediment and boulder distribution maps formed the basis of the subsequent BHT, OHT, and HUB maps, which were created using ArcGIS Desktop 10.7. All maps can be found in the Appendix A and are available for download as an ArcMap package.

### 2.2. Compiling the BHT Map (Benthic Broad Habitat Types according to EU Commission Decision 2017/848/EU)

The BHT sediment definition according to EUNIS in the area of the German Baltic Sea differentiates types of the infralittoral (light-flooded) and the circalittoral (nonlight-flooded) zones such as “mud”, “sand”, “coarse sediment”, “mixed sediment”, and “rocks and biogenic reef”. “Mixed sediment” corresponds to a hard substrate fraction of 10–90 % cover. The category of “rocks and biogenic reef” only refers to the occurrence of biogenic reefs because of the lack of information on specific coverages of geogenic hard substrates. A distinction between above 90% (“rocks and biogenic reef”) and below 90% hard substrate (“mixed sediment”) could thus not be made. However, it cannot be ruled out that the hard substrate cover locally exceeds 90 %. Only peat bottoms entered the category as biogenic reefs because they were covered by mussels. Geogenic hard substrates (categorised as “mixed sediment” in this study) within the detail areas is assigned when >5 boulders are present in a 50 × 50 m cell, or if both 1–5 boulders (as seen from the boulder distribution map) as well as lag sediment (as seen from the sediment distribution map) occur in a cell. Otherwise, the sediment is defined as soft substrate. In addition to the sediment map of Tauber [32], the hard substrate was assigned according to the reef coverage, which, for the first time, was delineated over a large area for reef designation according to [8] within the detail areas. These areas were reported to HOLAS III (HELCOM holistic assessment). Outside of the detail areas, reef coverage in Schleswig-Holstein [38], Mecklenburg-Western Pomerania [39], and the EEZ (BfN) was used for further hard substrate allocation. Mapped reef areas and suspected reef areas were assigned to the BHT “mixed sediment (hard substrate)”, and the polygon areas were gridded. Reefs in Mecklenburg-Western Pomerania, in contrast to those in Schleswig-Holstein (at 50 × 50 m), were shown at 1 × 1 km because they were not mapped out. The sediment map of Tauber [32] was used for the soft substrate allocation outside the detail areas. An overview of the data basis entered for the BHT, OHT, and HELCOM HUB maps can be found in Table 2.

The sediment classification schemes used for hydroacoustic interpretation and by Tauber [32] are incompatible with the biotope classification systems; therefore, a translation was necessary. The reclassification for Tauber is documented in Table 3. The silt and gravel/coarse sand content of each sediment class from the map according to Tauber was used from the underlying sediment analyses to categorise the sediment classes according to the definitions by EUNIS. Habitat categorisation according to EUNIS [43] is defined as “muddy sediment” if the mud, silt, or clay (<63 µm) content is at least 20%; “coarse sediment” if the mud/silt/clay fraction is less than 20% and the gravel and pebbles (2–63 µm) exceed 30% of the combined gravel and sand fraction; and “sand” if the mud/silt/clay fraction is less than 20% and the sand (0.063–2 mm) exceeds 70% of the combined gravel and sand fraction. The categories “muddy sediment”, “coarse sediment”, “sand”, or “rock and boulders” (>63 mm) are used when a coverage of at least 90 % is reached. “Mixed sediment” is used if the coverage of hard (rock/boulders/stone) and soft substrata (muddy sediment/sand/coarse sediment) is at least 10–90%. EUNIS is therefore based on the HELCOM HUB classification. The silt fraction (grain size < 63 µm) of the sediment class “very fine sand” from the map according to Tauber, for example, was >20% and was therefore assigned to the EUNIS/BHT type “mud”. As the sediment characteristics varied considerably in the detail areas, reclassification from the hydroacoustic surveys to EUNIS sediment types was carried out in an area-specific manner.

For photic zonation, a modelled polygon shape from [44] was used. Photic zonation was assigned in this study to the infralittoral zone, and aphotic zonation was assigned to the circalittoral zone.

For this study, BHTs occupying an area of <1 ha were eliminated and aligned with the surrounding BHTs. Areas within the inner coastal waters were retained as shown in the latest BHT map prepared in 2018 [45].

The nationally protected habitat type involving “species-rich areas of gravel, coarse-sand and shell-gravel areas” (§30 of the German Federal Nature Conservation Act) was specified as BHT “sand” rather than “coarse sediment” because the condition of >30% gravel or coarse sand content (EUNIS) was not met, based on the available sediment distribution maps.

### 2.3. Compiling the OHT Map (Benthic Other Habitat Types according to EU Commission Decision 2017/848/EU)

In contrast to BHTs, OHTs were included in the map as they were. For example, if reef areas were reported as polygons nationally, this area was integrated into the map as it was, and not gridded. Reefs mapped in this study were derived according to the national guideline for large-scale delineation of geogenic reefs in the German Baltic Sea [8]. The guideline specifies certain rules for gap closure within reef occurrences, so that the delineated reefs do not have to completely match with the indication of the BHT “mixed sediment”. Thus, in a cell where habitat type 1170 is indicated, the BHT “sand” may occur.

Sandbanks were shown in the same manner as the reefs from Schleswig-Holstein (Schleswig-Holstein State Office for the Environment, LFU [38]), Mecklenburg-Western Pomerania (Leibniz-Institute for Baltic Sea Research Warnemünde, IOW; State Office for the Environment, Nature Conservation and Geology Mecklenburg-Western Pomerania, LUNG [39]) and the EEZ (IOW; Christian-Albrechts-University Kiel, CAU Kiel; Federal Agency for Nature Conservation, BfN; Federal Maritime and Hydrographic Agency, BSH). Sandbanks in “Fehmarn Belt” and “Adler Ground”, as identified and described in the study by Boedeker et al. [46], were remapped by CAU Kiel and IOW and intersected with the reef cover from 2022.

“Seagrass meadows and other marine macrophyte populations” (§30 Federal Nature Conservation Act) and “Baltic aphotic muddy sediment dominated by ocean quahog (*Arctica islandica*)” (HELCOM Red List) were modelled in this study in contrast to the other OHTs as HELCOM HUB biotope type and integrated into the OHT map (see Section 2.6 and Section 2.7). *Zostera* spp. and *Fucus* spp. distribution areas for Schleswig-Holstein ([40]; data from the State Office for the Environment Schleswig-Holstein, 21 February 2022 and 18 March 2022) that were already modelled and mapped eelgrass beds for Mecklenburg-Western Pomerania [41] were integrated into the modelled HUB map at the end and indicated as OHT. *Fucus* spp. or *Zostera* spp. entered a cell as soon as they were modelled with an occurrence of at least 50 % (this also corresponds to the prediction probability) or mapped with at least 10 individuals/m^2^. The biotope type “seagrass meadows and other marine macrophyte populations” also includes foliose and corticated red algae, which were not indicated here in favour of the reef indication as habitat type, except when individual occurrences were observed outside the reef cover.

### 2.4. Biological Mapping

A total of 1637 grab samples, 403 station videos, 59 station photos, and 47 photo transects were taken and processed. Dominant benthic communities were classified in preparation for the HELCOM HUB map. Data from grab samples were used to determine dominant endobenthic organisms, and video and photographic records as well as diver samples were used to determine dominant epibenthic organisms.

Grab sampling was conducted using a Van Veen grab (0.1 m^2^) (Alu-Bau Ltd, Büdelsdorf, Germany) with an additional sediment sample obtained for granulometric analysis. The benthic samples were flushed through a sieve with a mesh size of 1 mm or, in the case of coarser sediment content, suspended in several subsamples, and the supernatant was decanted and poured through a 1 mm sieve again. The sample was fixed using a 4% formalin buffer solution, and marble grit was added to preserve mussel and snail shells. In the laboratory, the specimens were determined to species level, if possible, using a Carl Zeiss Discovery.V8 binocular (Carl Zeiss AG, Oberkochen, Germany). The wet weight was determined. Determination of dry weight and ash-free dry weight was carried out using Leibniz Institute for Baltic Sea Research (IOW) internal conversion factors [47]. Sampling was carried out according to standard instructions [48,49].

In addition to grab sampling, optical methods (underwater video and photography) were used to record epibenthic colonisation. Simultaneously with grab sampling, video recordings were taken at the grab station sites using a SeaViewer Sea Drop 6000 HD for a minimum of 5 min. For transects (0.3–2 nm), the recordings were obtained using a towed camera system developed at IOW (BaSIS—Baltic Sea Imaging System [50]), towed at ~0.5 kn. This camera system took one image every 15 s, of which one photo per minute that was suitable for analysis (not blurred, no shadows, and no sediment turbulence) was selected. In addition, in one campaign, an external drop camera system was used in the EEZ, which was designed by the German Federal Agency for Nature Conservation (BfN). This drop camera frame, equipped with a GoPro HERO4 Black, took a picture every 5 s at a station.

Video analysis (SeaViewer) was semi-quantitative based on the estimated coverage of epibenthic taxa and substrate according to the ACFOR scale (abundant, common, frequent, occasional, rare), which was visible in 5 min of video recording at the station. Image analysis (BaSIS, BfN drop camera) was performed quantitatively using the open-source software CoralPhotoCount 4.1 with an Excel extension (CPCe [51]), as described in the study by Beisiegel [50].

Furthermore, diver sampling was performed, during which scratch samples were obtained through collection frames (0.1 m^2^) with attached net bags, where the surface of a stone/boulder was scraped off within the frame, and biomass (dry weight) was determined. In addition, diver photos were used to estimate the degree of coverage of the epibenthic organisms.

Dominant benthic communities were indicated according to the HUB (HELCOM Underwater Biotope and Habitat Classification) system published by HELCOM [9]. Individual HUB classes were assigned manually at each station or georeferenced transect section (still images). As a result, several HUB classes in one cell could be included (as response variables) in the modelling. Both soft- and hard-bottom classifications were carried out separately. Endobenthos classification was assigned first from grab samples, and then epibenthos classification was assigned from video and photographs. For an endobenthic taxon, the critical value was based on a biomass fraction of >50%, and for an epibenthic taxon, the criteria were based on coverage of 10% on the total area or 90% on hard substrate to be considered dominant [9]. Assigned HUB classes were then represented areally by predictive habitat modelling (see Section 2.6).

### 2.5. Data Basis for Modelling

In addition to the data collected in the current study’s projects and data from the IOW database, acquired data from the 2010–2021 period were used (~45% internal and ~55% external data). Table 4 shows the amount of data and where it was derived from. These data originate from grab samples provided by authorities and private sector service companies (LUNG, LFU, StALU MM, StALU WM, WSA Stralsund, IfAÖ, Palaemon aquatic service company). A total of 3,628 stations were included in the model for endobenthic communities (Figure 2).

External data used for sessile epibenthos modelling came from photo-recorded diver sampling from management plans [52,53,54,55,56] and diver scratch sampling [57,58]. A total of 3623 stations and transect sections from the 2010–2021 period (~92% internal and ~8% external data) were included in the model (Figure 3). All cells with a larger areal proportion of hard sediment (>5 boulders or lag sediment with at least one boulder per 50 × 50 m cell) to soft sediment within a 1 × 1 km cell were included in the epibenthos modelling. The basis for the hard bottom modelling was the current reef boundaries of the coastal waters of Schleswig-Holstein, Mecklenburg-Western Pomerania, and the EEZ. In the area of Schleswig-Holstein, this includes suspected reef areas and geologically as well as biologically verified reefs [38]. In the area of Mecklenburg-Western Pomerania, the data were derived from the management plans and the suspected habitat type areas (according to the Habitats Directive) from 2011 [39]. In the EEZ, it consisted of reef areas mapped by CAU Kiel, BSH, and IOW through the EEZ project 6 and the project SEDINO phases I, II, and III (both funded by BfN).

### 2.6. Predictive Biotope Modelling

The HELCOM HUB map was created using predictive habitat modelling, unlike the BHT and OHT maps (except for HELCOM HUB biotope types included therein). In preparation for the modelling and subsequent HUB biotope map, a grid of 1 × 1 km grid cells (corresponding to the EEA standard grid) was placed over the coastal waters and EEZ, with each cell assigned a unique entry from the abiotic variables. If a grid cell contained multiple sediment types, the sediment with the higher proportion within the cell was assigned to the cell. The same procedure was used for the detail areas with a 50 × 50 m grid. Both the overview area and the detail areas were each modelled separately. The data used for the detail areas also went into the modelling of the overview map.

A random forest classification model (after [59]) was used to predict HUB biotopes. Modelling was carried out separately for endobenthos and epibenthos using the “randomForest” package (version 4.6–14, [60]) in RStudio 2022.12.0 (R environment version 4.2.2, the R Foundation for Statistical Computing Platform).

First, with respect to modelling, the already assigned HUB classes of each station/transect section (as described in Section 2.4) were specified at levels 4–6 (biotope level, without sediment information), and after modelling, the predicted HUB codes were completed with the found sediment and photic zone in the respective cell (levels 1–3) according to the definitions of HELCOM (for the HUB map). The previous manually assigned HUB classes entered the model as response variables and were used to classify HUB classes in every cell of the German marine waters (for endobenthos) and the hard substrate areas (for epibenthos). This study therefore follows a community-based modelling approach, as described by other authors [61,62,63]. In addition to the soft-bottom data from the sediment distribution maps of the detail areas, the following raster datasets from both the ERGOM model (Ecological Regional Ocean Model, model run from 2010 to 2017, [64]) and the GETM model (General Estuarine Transport Model, model run from 2010 to 2020, [65]) were available as predictors at a 600 × 600 m resolution:Temperature, salinity, current velocity (in directions of north/south, east/west, without directional information), and bottom shear stress from the GETM model [65];Photosynthetically active radiation (PAR), oxygen concentration, number of hypoxic days, DOC, ammonium, nitrate, phosphate from the ERGOM model [64];Water depth and sediment type [32];Photic zonation (based on ERGOM model, [44]);Slope gradient (based on [32]).

The values from these raster datasets were assigned from the centre point of a cell. Outside of the detail areas, where no areal geologic mapping was conducted, the sediment map of Tauber [32] was used as a predictor for soft-bottom categorisation. Slope was only included in the epibenthos modelling and was created from the bathymetric map of Tauber [32] using the Spatial Analyst tool in ArcGIS Desktop 10.7.1. PAR (photosynthetically active radiation), and photic zonation was also included in the epibenthos modelling only. The polygon shapefile used for photic zonation is based on the light penetration depth (PAR) values from the 2000–2010 ERGOM model run [44]. To separate the photic and aphotic zones, the 1% light penetration depth (averaged over the growing season from March to October) was coupled with bathymetry [66]. The initial dataset was randomly divided into a training dataset (70%) and testing dataset (30%). To improve model performance, hyperparameters (number of trees and number of predictors at each decision node) were tuned until lowest out-of-bag (OOB) error was found, and model adjustments were made if the dataset was imbalanced (using downsampling, balanced random forest, upsampling, and the SMOTE algorithm).

### 2.7. HUB Map Modelling Limitations and Conventions

In general, the model performance decreases when modelling classes are very similar to each other, for example, when separating and predicting a biotope class of a dominant specific species from a biotope class of a community containing exactly the same species. Therefore, the following conventions had to be adopted in the modelling process (based on Section 2.5 and Section 2.6):Elimination of outliers:Before modelling the endobenthos in the whole German Baltic Sea, stations dominated by taxa that rarely occurred in the area and that accounted for max. 1% of the total number of stations were eliminated. Such outliers were Actiniaria and oligochaetes (in HELCOM HUB they are classified as meiofauna).*Ophelia* spp./*Travisia* spp. could not be separated from other communities by the random forest (RF) model and therefore were not reliably predicted, so stations with dominant *Ophelia* spp./*Travisia* spp. were also deleted.Less frequent dominant taxa were assigned to a higher category:Dominant *Mya arenaria* and *Astarte* spp. were assigned to the community with multiple infaunal bivalve species, because being a part of the overarching community, they were poorly separable from each other. Because the polychaete communities (partly with dominating *Scoloplos armiger*, *Marenzelleria* spp., *Pygospio elegans*, and *Hediste diversicolor*) were difficult to separate from the other communities; they were grouped together as the community with macroscopic infaunal biotic structures (HUB Level 4), as were stations ending at HUB level 5 (e.g., dominant bivalves/polychaetes/crustaceans). Therefore, the community with macroscopic infaunal biotic structures includes not only communities without dominant taxa, but also those previously mentioned that are too unspecific in their occurrence, leading to improved model performance.Non-dominant communities were indicated as dominant:Epibenthos-dominated stations that ended up at HUB level 5 were indicated as HUB level 6 (e.g., foliose red algae were treated as dominant even though they had < 50% cover), because the model cannot separate dominant and non-dominant communities, in order for those stations to be included in the model. This means that in areas where epibenthic communities are predicted, they do not need to be dominant, but they are more likely to occur than other communities.Mixed communities were indicated as non-mixed communities:Mixed communities that are very similar in species composition (e.g., foliose red algae, foliose red algae/sponges, foliose red algae/filamentous red algae, foliose red algae/bryozoans, and foliose red algae/sponges/kelp) cannot be clearly delineated by the model. Therefore, these mixed communities were assigned to those taxa that play a superior role in the biotope function (structuring, long-lived, and geographically dominant). For example, the classes listed above were assigned to dominant foliose red algae. This means that epibenthic mixed communities can always occur, even when indicated otherwise. Transitions cannot be modelled with the procedure chosen here because the model considers each class as distinct.

The predictions of endobenthos and epibenthos from the models were intersected eventually, in the sense that the epibenthic community in a cell was indicated at the sites where hard substrate dominates. Unlike the BHT map, the HUB map was not generalised (i.e., areas < 1 ha were not matched to surrounding sediment).

Benthic broad habitat types and other habitat types according to the Commission Decision [11] are aligned with HELCOM HUB biotope types. This means that habitats (i.e., sediment information) coincide, except for the indication of OHT “reefs” and BHT “mixed sediment”, as different delimitation rules underlie here (see Section 2.3).

## 3. Results

### 3.1. Broad Habitat Type (BHT) Map

The map in Figure 4 shows the broad habitat types at a 1 × 1 km resolution with the incorporation of the areas mapped at a 50 × 50 m resolution in this study (“Outer Wismar Bay”, “Darss Sill”, and “Plantagenet Ground” in the coastal waters of Mecklenburg-Western Pomerania, as well as the nature conservation areas of the EEZ, “Fehmarn Belt” and “Kadet Trench”). Infralittoral sand and circalittoral mud occupy the largest areas in the German Baltic Sea, with each being >20% of the total area (Table 5), followed by circalittoral sand, infralittoral mixed sediment (hard substrate), infralittoral mud, and circalittoral mixed sediment (hard substrate). The remaining BHT categories amount to less than 1% of the total area.

Major differences to the previous version of the map from 2018 [45] are the update of hard-bottom areas and the detailed representations of sediment compositions in selected areas. However, the assignment of sediment types (Table 3) shown according to [32] also differs from the sediment reclassification in the map submitted to HOLAS II; for example, mudflats (e.g., west of Fehmarn, in the Plantagenet Ground, east of the Isle of Rugia) are more widespread or larger than in the 2018 map. Another difference is the photic zonation. While a layer from the EUSeaMap was used for the previous map, a more detailed shapefile from [44] was used here for the classification into infralittoral (photic) and circalittoral (aphotic) zones. The area of the infralittoral zone is larger in the shapefile used in this study, with the Kiel Bight, in particular, differing on a large scale, and the rest differing on a rather small scale. The inner coastal waters are consistent with the 2018 map. The only change was made in the Szczecin Lagoon, where circalittoral mud and sand were changed to infralittoral mud and sand.

### 3.2. Other Habitat Type (OHT) Map

Reefs occupy the largest area of all OHTs with 2183.5 km^2^ (Figure 5, Table 6). They consist mainly of boulder fields and extend mostly on abrasion platforms that continuously expose boulders during the ongoing erosion of glacial till [34]. So far, only small areas of pure lag sediment reefs have been mapped. Biogenic reefs (pure mussel beds) have not yet been observed.

The nationally protected habitat type involving “species-rich areas of gravel, coarse-sand and shell-gravel areas” consists of a suspected area found in the detail area of the “Darss Sill”. The total area covered is 9 km^2^. Reef areas and “species-rich areas of gravel, coarse-sand and shell-gravel areas” partly overlap. However, since both habitat types cannot be designated as protected biotopes at the same time, the area is reduced to almost 6 km^2^.

“Seagrass meadows and other marine macrophyte populations” (§30 Federal Nature Conservation Act) containing modelled eelgrass is found near the coast in the light-flooded areas. However, they have not yet been mapped for the inner coastal waters of Mecklenburg-Western Pomerania. The only mapped seagrass meadow is located in the “Plantagenet Ground”. Seagrass occurrences were recorded using hydroacoustic data (side scan sonar) and could be verified using video footage. They occur in the east of the detail area on fine sand. The stock thins out to the north. Delineation to the 10 % cover is not possible using side-scan sonar mosaics due to shadow formation. Since very shallow areas could not be approached by the research vessel, it is uncertain whether this nationally protected biotope type extends over a larger area towards the west. Seaweeds (*Fucus* spp.) occur more frequently in denser populations in the coastal waters of Schleswig-Holstein. Foliose red algae, such as *Delesseria sanguinea*, occur in marine areas approximately as far as the Darss Sill and where more saline water can flow through the Kadet Trench to the east. They largely dominate the reefs, but are not specified in favour of habitat type 1170. Corticated red algae, such as *Furcellaria lumbricalis*, could only be mapped sporadically as the dominant stock (see also Section 3.3).

“Baltic aphotic muddy sediment dominated by ocean quahog (*Arctica islandica*)” are found in deep basin areas where currents are low enough to allow for fine sediments to deposit, such as Eckernförde Bay, Mecklenburg Bay, and Arkona Basin. It occupies the second largest area of the OHT with 1417.6 km^2^.

### 3.3. HELCOM HUB Map

A total of 84 HUB biotope types could be modelled in the detail areas and the entire German Baltic Sea (Table 7). Figure 6 shows the HUB biotope map for the German Baltic Sea, and Figure 7, Figure 8 and Figure 9 show the HUB biotope maps for the detail areas in Mecklenburg-Western Pomerania (for more information on the detail areas in the EEZ, see [26,27]). The colours represent sediment types and the shadings represent benthic communities. NAs result from non-evaluable data in the boulder distribution maps and from sediment types that were not included in the model (due to missing benthological ground truthing) and therefore could not be predicted. However, this accounts for only about 8 km^2^ in the detail areas of “Kadet Trench”, “Outer Wismar Bay”, “Darss Sill”, and “Plantagenet Ground”.

The selected models for the detail areas and the overall area are shown in Table 8. The modelling of the endobenthic communities in the areas of “Plantagenet Ground”, “Kadet Trench”, and “Fehmarn Belt” achieved a higher model goodness of fit (AUC = 0.79–0.8) than those for the coastal areas, “Outer Wismar Bay” and “Darss Sill” (AUC = 0.65–0.76), where the biotope classes were more difficult to distinguish from each other. The values for the overall German Baltic Sea model were in the middle range (AUC = 0.70). The results of the modelling of the epibenthic communities showed a very high model goodness of fit (AUC > 0.9) for the “Outer Wismar Bay”, the “Darss Sill”, and the “Fehmarn Belt” areas. In contrast, the values of the “Kadet Trench” (AUC = 0.71) and the overall area (AUC = 0.81) were lower. The epibenthos in the “Plantagenet Ground” was not modelled, as only mussels were observed on the hard substrate in the entire area.

The wide sandy areas in the Pomeranian Bay and the Rugia-Falster Plateau are colonised by multiple infaunal bivalve species (consisting of *Cerastoderma glaucum*, *Macoma balthica*, *Mya arenaria*, *Astarte borealis*, and *Arctica islandica*). Silty areas in the Arkona Basin are particularly dominated by *Macoma balthica*, which, although also a component of the aforementioned community, is the main dominant species, especially in this area. Other basins where mud is deposited, such as the Mecklenburg Bay, the Fehmarn Belt, the Eckernförde Bay, and parts of the Arkona Basin, are dominated by *Arctica islandica*. Especially in the first two areas, the prediction confidence that ocean quahog is dominant is high (>80%). Mussels (also as part of the endobenthos) are correctly predicted where reef structures or hard bottoms are present. The habitats characterised by macroscopic infaunal biotic structures (ending on HUB level 4) covers not only communities where no taxa dominate, but also polychaete communities, and generally dominant bivalves and crustaceans. Particularly, in the areas where the sediment is heterogeneously distributed on a small scale (nearshore areas off northwest Mecklenburg and the Rostock district, the Darss Sill, and the coastal waters of Schleswig-Holstein) or due to the lack of data in the nearshore areas in Schleswig-Holstein or in the southwestern Arkona Sea, the prediction probability of the model is low (<50%).

The biotope map shows that not only does sediment influence the spatial distribution of benthic communities, but also salinity, which is observable at the Darss Sill, which is a natural barrier. In front of it (in the western Baltic Sea), a wide variety of marine communities occur, whilst behind it, specialists adapted to brackish water have established themselves. The salinity gradient is also visible in the spatial distribution of the epibenthic communities (see also Table 8). While there are still numerous mixed communities of various colonisers off the Darss Sill, the number of species decreases steadily towards the east. In the Bay of Kiel, communities with non-filamentous corticated red algae, such as *Furcellaria lumbricalis*, dominate the coastal waters. Towards Fehmarn and Mecklenburg Bight up to the Kadet Trench, predominant communities are foliose red algae and mussels, while the deeper, poorly lit areas are colonised only by hydrozoans or are sparsely colonised. In the eastern Baltic Sea, mainly *Mytilus edulis* communities dominate the hard substrates. Mixed communities are rarely found here anymore.

In the “Plantagenet Ground”, an area of peat with a thin sand layer was identified using a video transect (Figure 9). It is colonised by mussels (with a cover of filamentous algae) and was therefore classified as peat bottom with mussels on sand (AA.G+AA.J1E1). Thus, peat is also considered a reef-building substrate.

Benthic communities modelled in both the study by Schiele et al. [20] and the current study show similar spatial distributions. This is found for *Arctica islandica,* the multiple infaunal bivalve community (HUB code L9), the Mytilidae community, and *Macoma balthica*, whereby the latter’s distribution range extends further south according to Schiele et al. [20] than in this current map. However, with *Macoma balthica* being the dominant species and also occurring within the (L9) community adjacent in the south, the boundaries of its distribution range are likely to be fluid. A difference in the degree of detail between the two maps is further evident at the outer edge of the Arkona Basin on the German side, where Schiele et al. [20] indicate Bivalvia (ending at HUB level 5), whilst here, the multiple infaunal bivalve community, the *Arctica islandica* community, and macroscopic infaunal biotic structures (ending at HUB level 4), respectively, were modelled. The reef structure east of the Bay of Greifswald was mapped only after the publication of the 2015 biotope map, so that mussel occurrence increases here.

Another significant difference is that additional epibenthic communities, such as red algae, hydrozoans, barnacles, and moss animals, were modelled here, and *Zostera* spp. and *Fucus* spp. were added. Due to the fact that reef structures were further mapped and sampled after 2015, additional epibenthic biotope types, including sparse (2T) and non-existent colonisation (4U), could be indicated here.

Macrophytes or algae were not differentiated into perennial or annual macrophytes in the study by Schiele et al. [20], whereas in this study, perennial macrophytes and algae were differentiated. This provides a more accurate picture of the occurrence of specific morpho-species. Nevertheless, the distribution areas are similar. Furthermore, it should be mentioned that annual filamentous algae are often associated with other taxa, and such mixed communities were assigned to another taxon at the expense of the algae in this work. This means, e.g., that when annual filamentous algae co-occurred with mussels, the biotope was assigned to mussel-dominated areas. Thus, depending on the season, annual filamentous algae can also be found more widely distributed than shown in this HUB map (Figure 6).

The HUB, BHT, and OHT maps are largely congruent. The photic zonation and substrate allocation are the same for the HUB and BHT map. The OHT map partly includes results of the HUB modelling (“seagrass meadows and other marine macrophyte populations” and “Baltic aphotic muddy sediment dominated by ocean quahog (*Arctica islandica*)”). Sandbank areas are also shown as “sand” in the HUB and BHT maps. Only “species-rich areas of gravel, coarse-sand and shell-gravel areas” are indicated as “sand” in the BHT and HUB maps, as this substrate does not correspond to the EUNIS or HUB type “coarse sediment” according to the sediment analysis, but to “sand”, as already described in Section 2.2. Other exceptions are the reef areas, which do not fully correspond to the BHT/HUB substrate “mixed sediment”, as described in Section 2.3.

## 4. Discussion

The new maps now integrate the latest mapping results of widespread habitats, their benthic communities, and protected habitats and biotope types in one map package. Due to the, in some part, high-resolution, standardised, up-to-date mapping and improved modelling through a larger data basis, a more precise picture of the state of the seafloor in the German Baltic Sea is now provided (Figure 10). The map according to Tauber [32] is a pure soft-bottom sediment map interpolated from a large dataset of grab samples. A separate hard bottom map created by the same author only gives roughly drawn polygons from point observations [32]. Although the sediment information gives a correct representation of the seabed when viewed over a large scale, it is too inaccurate when viewed over a small scale. The habitat maps from the current study can now replace the previous sediment and boulder maps according to Tauber [32] in selected areas (two nature conservation areas in the German EEZ and three detail areas in Mecklenburg-Western Pomerania) as well as the biotope map of Schiele et al. [20]. This is due to an improved methodology and more recent mapping results, which increase the level of detail compared to earlier maps, and it should not be interpreted as an indicator of temporal habitat change. [20] used a dataset from 2004 to 2013, while in this modelling, data from 2010 to 2021 were used, partially overlapping with Schiele’s dataset. The temporal factor (as well as seasonality) was neglected in this work, as the focus was on the spatial distribution of habitats and biotopes. For a more accurate assessment of potential habitat changes, it is recommended to conduct precise mapping (using hydroacoustics and ground truthing) and delineation of an area already known and, ideally, captured according to national standards (such as habitat types according to the HD). The maps serve as the basis for this purpose.

### 4.1. Modelling Biotope Distributions

Benthic communities settle on certain substrate types under specific conditions of salinity, light availability, exposure, etc. [67,68,69], which was reflected in the importance of variables in our model building, where parameters such as sediment, salinity, and depth played major roles, especially at large scales (Table 8). On the other hand, at small scales, the most important predictors were the dissolved organic carbon content (“Darss Sill”, “Outer Wismar Bay”, and “Fehmarn Belt”), sediment (“Plantagenet Ground” and “Fehmarn Belt”), current velocity (“Outer Wismar Bay” and “Kadet Trench”), depth (“Fehmarn Belt”), bottom temperature (“Darss Sill”), oxygen content (“Outer Wismar Bay”), and bottom shear stress (“Kadet Trench”). Small-scale processes, which can overlap large-scale ones, are relevant in the detail areas, which are reflected in the formation of different biocenosis. For example, in the channel system of the “Kadet Trench”, the bottom shear stress and current velocity play major roles in the distribution of endobenthic communities within the channels or on the flanks and reef flats. However, both parameters are also related to water depth and sediment. In the shallower reef areas, the currents reach a higher velocity than in the deeper channels. Fine sediment is washed away from the lag sediment areas above and deposited within the deep channels, where the current velocity decreases. On these fine sediments, a multiple infaunal bivalve species community settles with *Arctica islandica* or *Macoma balthica* dominating in certain areas, respectively. This community was easily distinguishable for the random forest model from the community characterised by macroscopic infaunal biotic structures predicted in the shallower areas with higher current velocity, following the simulated current velocity by the GETM model [65]. The interaction of several environmental parameters defines the benthic community formation. Large-scale gradients such as salinity, which cause a shift in benthic community composition in the Baltic Sea [2,70] are replaced on a small scale by other environmental factors that influence the diversity of community structures through their local heterogeneity [71,72].

In addition to model statistics (Table 8), prediction probabilities (not shown here) and the resulting biotope maps also determined model selection. A model was chosen if the biotope classes were predicted where they were actually found, and the biotope map generally showed high confidence (at least a 67% prediction probability). The dominant taxa were superimposed on the biotope maps for additional validation and compared. With very few exceptions, the dominant taxa found in ground truthing coincided with those predicted by the model as biotope class. The “Plantagenet Ground”, which is located east of the Darss Sill barrier and where the number of marine species is thus strongly reduced due to reduced salinity, has a higher goodness of fit (endobenthos AUC = 0.80) with its very homogeneous sediment composition than the more diverse “Outer Wismar Bay” (endobenthos AUC = 0.76). Where the distribution of a community is limited by a boundary of divergent abiotic conditions, thus favouring biotope delimitation, biotope classes can be clearly distinguished from each other, as was the case at the “Darss Sill” (epibenthos AUC = 1). There, mussels dominate the southern part of the hard substrate, which is shallower with a higher temperature, lower salinity, and lower DOC content than the northerly deeper areas, where foliose red algae prevail. The extent of foliose red algae with a transition to filamentous red algae or hydrozoans in the “Outer Wismar Bay” also follows the simulated distribution of dissolved organic carbon from the ERGOM model and therefore reaches a high goodness of fit (AUC = 0.98). These degraded areas, where turbidity and organic sedimentation are high and oxygen depletion occurs, can only be successfully colonised by hydrozoans, whilst other epibenthic colonisers struggle to survive.

### 4.2. OHT “Species-Rich Areas of Gravel, Coarse-Sand and Shell-Gravel Areas”

The biotope type “species-rich areas of gravel, coarse-sand and shell-gravel areas” is present if, among others, the indicator organisms *Ophelia* spp./*Travisia forbesii* occur at three stations within an occurrence area, according to [42]. This condition was met for the sediment type gS-mxSa (gravelly sand to mixed sand) in the “Darss Sill” area where the biomass fraction of a single taxon was at least 10% or a combined biomass fraction (of both *Ophelia* spp./*Travisia forbesii*) comprising at least 5% of the total biomass. In addition, there were two stations on LagSed+mSa (lag sediment and medium sand), in close proximity to gS-mxSa, where both taxa occur. Furthermore, *Ophelia* spp./*Travisia forbesii* dominate at two stations on mxSa-gS, close to LagSed (lag sediment). Due to the similarity of mxSa-gS and gS-mxSa, both substrate types in the “Darss Sill” are considered as potential “species-rich areas of gravel, coarse-sand and shell-gravel areas”. However, both sediment types do not comprise >50 % of gravel, coarse sand, and shell fraction and therefore do not meet the conditions to be designated as “species-rich areas of gravel, coarse-sand and shell-gravel areas” [42]. Due to the heterogeneity of sediment composition during sampling, it is open to question whether thin covers of sand in this area are positionally stable or if they might instead cover coarse sediment within the suspected substrate types. Further, the area is in spatial proximity to lag sediment and is dominated by reef structures, which is considered an indication of the protected habitat type according to [42]. Towards the east/northeast, *Ophelia* spp./*Travisia forbesii* occurrence reaches into fine sand areas (with a transition to mixed and medium sand). Therefore, the high density of dominant indicator organisms generally supports the plausibility of this suspected area in the “Darss Sill” area.

### 4.3. Methodological Review

Methodologies differ per state for the external data obtained. Regarding the coastal waters of Schleswig-Holstein, it is above all the mapping of the epibenthic organisms that differs from the procedure at IOW (and thus also Mecklenburg-Western Pomerania). At IOW, mainly video and photo techniques are used, which are supported by diver sampling, while in Schleswig-Holstein, mainly diving is used. Therefore, there were differences in the data availability during the adaptation of these external data to our approach. In some cases, only biomass was taken, but no coverage was recorded. In order to preserve these data, the epibenthic colonisation shown in the biotope maps is therefore based on the coverage of video, photo, and diver samples, but also on the biomass fraction of the total biomass at a station. The external data, where only macrophytes were mapped as a part of MSFD monitoring, could unfortunately not be included in the epibenthic modelling.

Little benthos data were collected in the nearshore area, and the raster data of the predictors are also less reliable in this zone. For methodological reasons, this area has been included in the maps but should be treated with caution.

The random forest method used here is considered very well suited for biotope classification because it is a nonparametric, robust algorithm showing high performances in supervised machine learning methods [73,74,75] that can handle outliers and noisy or redundant input features [59,76]. The algorithm selects a variable out of a random subset of predictors that is most important for decision formation at each node of a branch of each decision tree [60]. Aggregating the outcome of many random trees leads to an increase in generalisation power [76]. The overall AUC (0.7–1) and Cohen’s Kappa values (0.5–1) indicated a good to excellent prediction [77,78]. Nevertheless, the endobenthos model of the “Outer Wismar Bay” only showed very poor performance regarding Cohen’s Kappa (0.035). However, a visual examination showed that even with the perceived poor model performance, the random forest algorithm provides biotope maps that can approximate reality.

The model also reaches its limits as the number of response variables (biotope classes) that are closely interrelated increases. Naturally, there are no strictly delineated sediment types, each with different biocenosis. Transitions of different grain size fractions are fluent, or sediment mixtures can cause benthic communities to overlap [67]. Particularly in the case of differently composed sedimentological substrates, high biodiversity can occur, where it becomes difficult for the Random Forest model to find patterns and boundaries, as was the case, e.g., in the “Outer Wismar Bay” area. However, this is a general difficulty in modelling and not a question of methodology.

Furthermore, an assemblage always consists of different community-associated species. The dominant species given here, which give a biotope its name, therefore always occur with other associated species. Thus, the maps do not lay claim to the exclusive occurrence of individual habitat-determining species, nor do these species occur with absolute confidence in certain areas, nor are these distribution areas fixed. Rigid boundaries, as the maps suggest, do not exist, and are instead fluid transitions. Sediment types, like their inhabitants, are subject to natural dynamics. Rather, the maps are intended to provide indications of the likely spatial distribution patterns of benthic communities at large and small scales, and they do not give any indication of the status of biotopes.

### 4.4. Outlook

The spatial extent of habitats and protected biotopes in high resolution is still unknown in vast areas of the Baltic Sea. The demand for biotope maps is increasing, so monitoring and mapping is ongoing and will continue, not only to detect spatial changes but also to detect temporal changes due to natural and anthropogenic causes. By observing habitat changes over time, possible habitat loss can be detected (with regard to MSFD Descriptor 6), impacts can be assessed, and measures can be taken. The interplay between applied and basic research can contribute to the direct implementation of nature conservation measures.

Comprehensive hydroacoustic mapping provides new insights, particularly in the area of suspect reef areas, which can now be identified more accurately and in much greater detail. This does not only fulfil national mapping requirements at high resolution, but also national reporting requirements (monitoring of spatio-temporal changes and improvement and restoration measures) for the implementation of relevant directives, as well as for marine spatial planning and specific projects. Particularly, in light of the forthcoming Nature Restoration Law, which includes the introduction of restoration measures for at least 20% of the EU’s marine and terrestrial ecosystems by 2030 and for all ecosystems in need of restoration by 2050, these maps are an important tool, e.g., for identifying potential restoration areas or assessing the success of measures. The comprehensive HUB map is also a valuable tool for monitoring and assessment under HELCOM and the MSFD, and for deriving targeted management measures. However, as the modelling of community distribution is highly dependent on the dataset used, the maps should be used with caution as a basis for detecting changes in biotope distribution and for projects in small-scaled areas.

## 5. Conclusions

For the first time, habitats and biotopes in the German Baltic Sea have been mapped at a level of detail that has not been available before. In this study, we mapped specific sediment types in their actual extent using side scan sonar and on a larger scale for the first time using neural networks for stone detection [23,24]. Furthermore, the latest mapping results from federal and state governments have been incorporated into the maps. Based on national guidelines that have been developed over the past seven years to standardise sediment and boulder distribution maps and the large-scale mapping of reefs [8,22], these maps have been improved and updated with the latest available data.

The spatial distributions of the protected biotope types here show that specifically, the geogenic reefs (HD, §30 Federal Nature Conservation Act) can now be exactly reported. The biological verification of these geogenic reefs was essential and paves the way for a subsequent designation of this protected habitat type at an official level. The same applies to the protected seagrass meadows (§30 Federal Nature Conservation Act) in the “Plantagenet Ground” and the sandbanks (HD, §30 Federal Nature Conservation Act) in the “Kadet Trench” and “Fehmarn Belt”. For the first time, a suspected area of the nationally protected habitat type “species-rich areas of gravel, coarse-sand and shell-gravel areas” was found and mapped in the “Darss Sill”.

## Figures and Tables

**Figure 1 biology-13-00006-f001:**
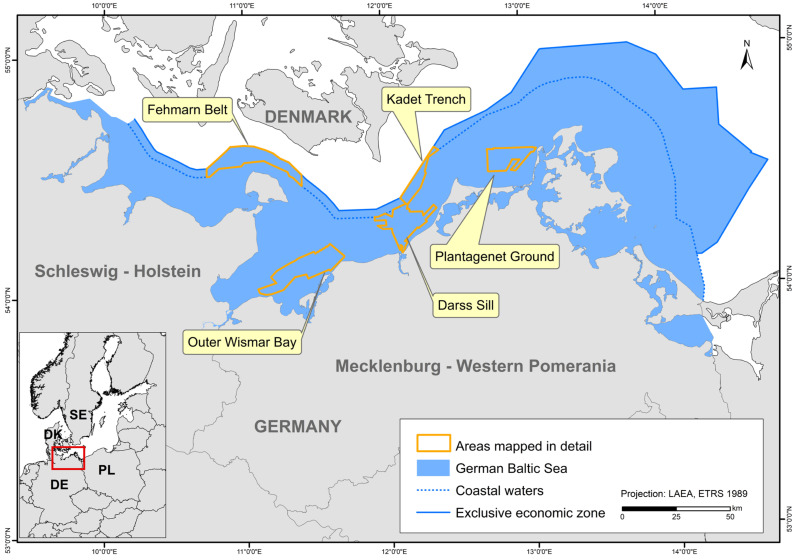
The German Baltic Sea including areas which were mapped in detail (resolution: 50 × 50 m).

**Figure 2 biology-13-00006-f002:**
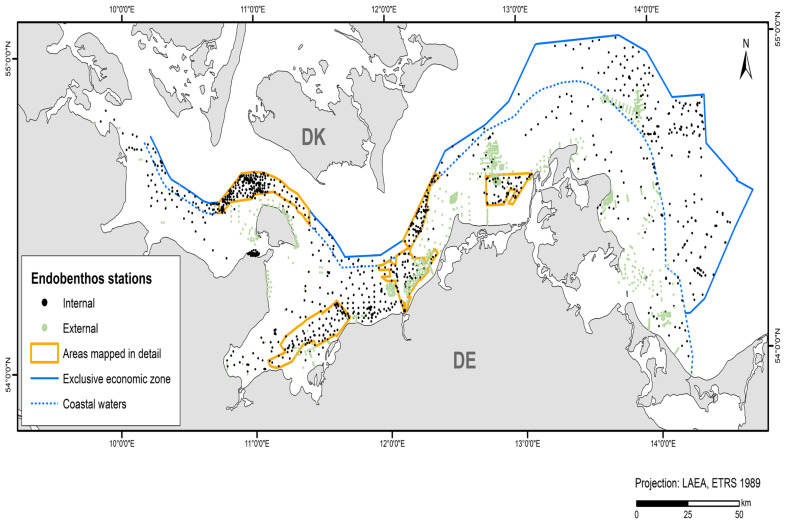
Stations and their data sources that entered the endobenthos model.

**Figure 3 biology-13-00006-f003:**
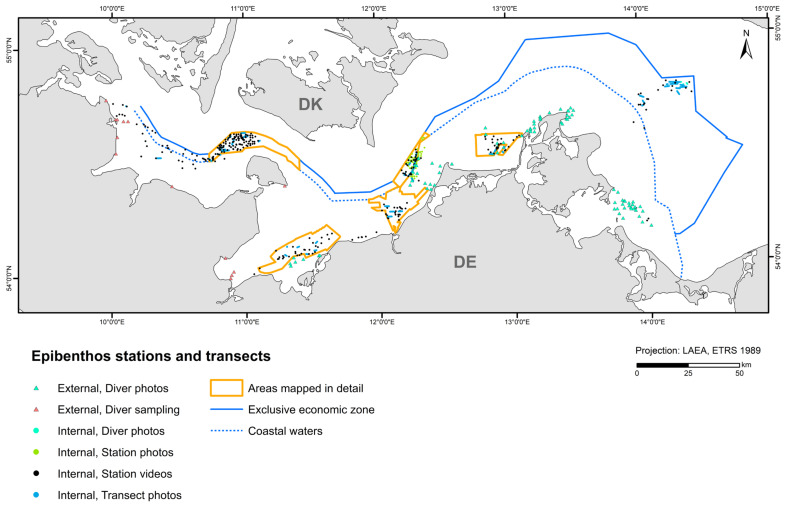
Stations and transects that entered the epibenthos model, indicating data source and type.

**Figure 4 biology-13-00006-f004:**
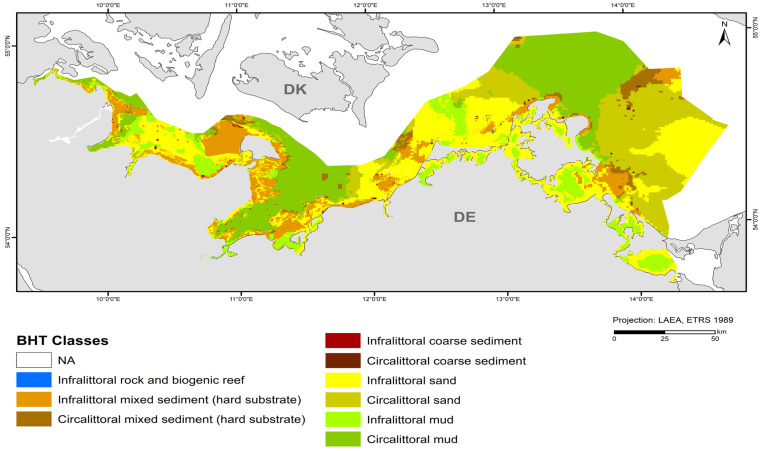
Distribution of broad habitat types (BHT) in the German Baltic Sea.

**Figure 5 biology-13-00006-f005:**
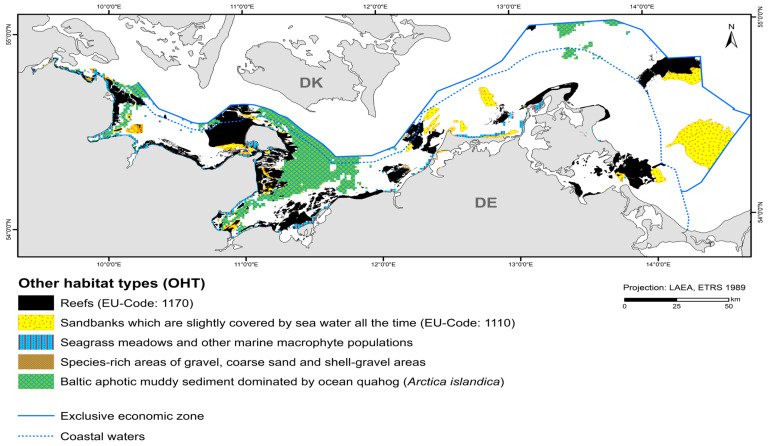
Distribution of other habitat types (OHT) in the German Baltic Sea that are protected under EU-/national law or included in the HELCOM Red List.

**Figure 6 biology-13-00006-f006:**
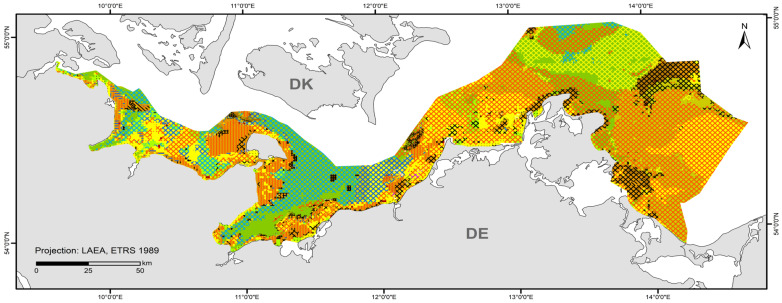
Distribution of HELCOM HUB Biotopes in the German Baltic Sea. For legend of colours, see Table 7.

**Figure 7 biology-13-00006-f007:**
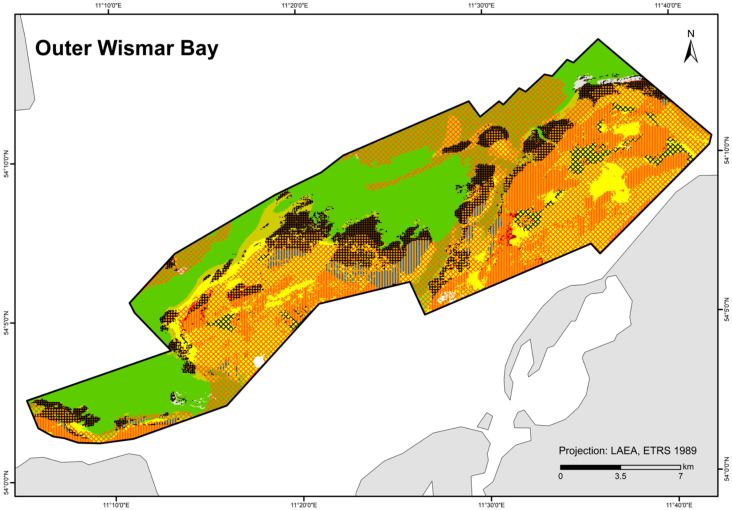
Distribution of HELCOM HUB biotopes in the “Outer Wismar Bay” area. For legend of colours, see Table 7.

**Figure 8 biology-13-00006-f008:**
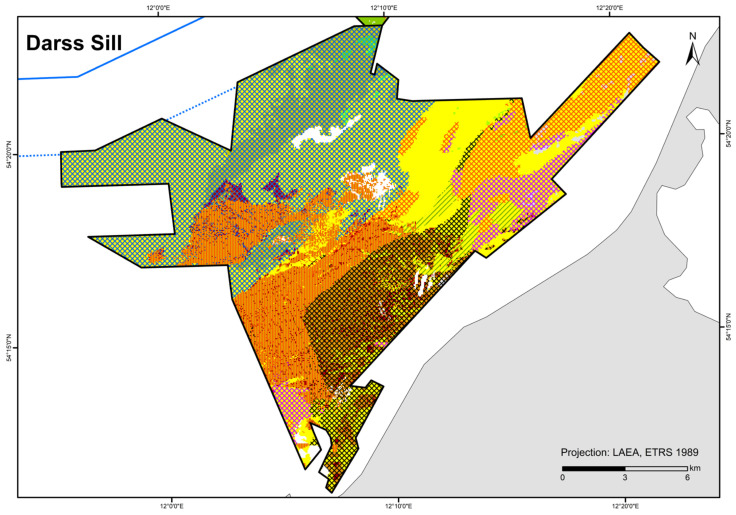
Distribution of HELCOM HUB biotopes in the “Darss Sill” area. For legend of colours, see Table 7.

**Figure 9 biology-13-00006-f009:**
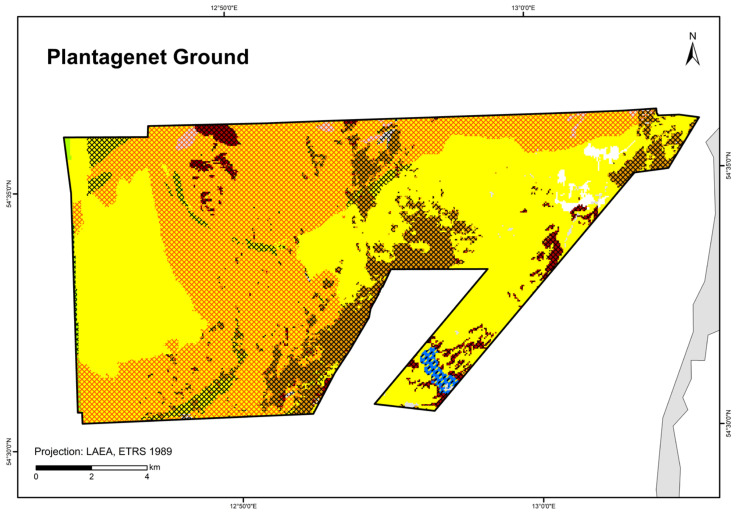
Distribution of HELCOM HUB biotopes in the “Plantagenet Ground” area. For legend of colours, see Table 7.

**Figure 10 biology-13-00006-f010:**
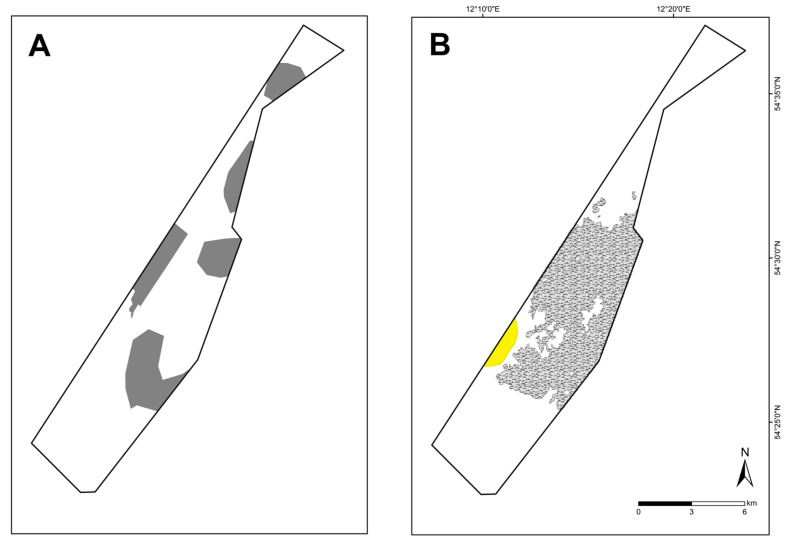
Previous suspected reef areas (**A**) and newly mapped OHT (**B**) reefs (grey) and sandbanks (yellow) in the Kadet Trench.

**Table 1 biology-13-00006-t001:** Level specifications of sediment classifications according to BSH [22]. * Not specified = lack of information and/or knowledge for exact classification. ** Not classified = cannot be classified further in this level.

Level A	Level B	Level C
Fine sediment (Fsed)	not specified *	not classified **
	mud (M)	not classified
	sandy mud (sM)	
	muddy sand (mS)	
Sand (S)	sand (S)	not classified
		fine sand (fSa)
		medium sand (mSa)
		mixed sand (mxSa)
		coarse sand (cSa)
Coarse sediment (Csed)	not specified	not classified
	gravelly sand (gS)	not classified
	sandy gravel (sG)	
	gravel (G)	
Mixed sediments (MxSed)	not specified	not classified
	gravelly mud (gM)	not classified
	gravelly muddy sand (msG)	
	muddy gravel (mG)	
Peat		
Lag sediment (LagSed)	not classified	not classified
Not specified	not specified	not specified

**Table 2 biology-13-00006-t002:** Data basis for the BHT, OHT, and HELCOM HUB maps inside and outside the detail areas. Note the distinction between endobenthos and epibenthos in the predictors used for HUB biotope modelling.

	BHT		OHT		HUB	
	Detail Areas	Outside of Detail Areas	Detail Areas	Outside of Detail Areas	Detail Areas	Outside of Detail Areas
Overall resolution	50 × 50 m	1 × 1 km	50 × 50 m and polygons	1 × 1 km and polygons	50 × 50 m	1 × 1 km
Map basis for soft bottom	Sediment distribution maps from hydroacoustic surveys (gridded)	Tauber [32] (gridded)	Seagrass meadows and “species-rich areas of gravel, coarse-sand and shell-gravel areas” mapped according to hydroacoustic results; distribution area of “Baltic aphotic muddy sediment dominated by ocean quahog (*Arctica islandica*)” modelled in this study	“Seagrass meadows” modelled by [40,41] (gridded); sandbanks as reported to HOLAS III (polygons); distribution area of “Baltic aphotic muddy sediment dominated by ocean quahog (*Arctica islandica*)” modelled in this study	Sediment distribution maps from hydroacoustic surveys (gridded)	Tauber [32] (gridded)
Map basis for hard bottom	Boulder distribution maps from hydroacoustic surveys according to [8] (grids)	Reef areas as reported to HOLAS III (gridded)	Distribution area of “other marine macrophyte populations” modelled in this study; reefs mapped hydroacoustically in this study (gridded)	Reef areas as reported to HOLAS III (polygons)	Boulder distribution maps from hydroacoustic surveys according to [8] (grids)	Reef areas as reported to HOLAS III (gridded)
Hard bottom assignment	>5 boulders/50 × 50 m cell or lag sediment and >1 boulder/50 × 50 m cell (from boulder and sediment distribution maps)	Reef areas as reported to HOLAS III (gridded)	Reefs mapped according to [8]	Reef areas as reported to HOLAS III	>5 boulders/50 × 50 m cell or lag sediment and >1 boulder/50 × 50 m cell (from boulder and sediment distribution maps)	Reef areas as reported to HOLAS III (gridded)
Biotope classification schemes	EUNIS	EUNIS	“Species-rich areas of gravel, coarse-sand and shell-gravel areas” according to [42]; “Seagrass meadows and other marine macrophyte populations” and “Baltic aphotic muddy sediment dominated by ocean quahog (*Arctica islandica*)” according to HUB; reefs according to [8]	“Seagrass meadows and other marine macrophyte populations” and “Baltic aphotic muddy sediment dominated by ocean quahog (*Arctica islandica*)” according to HUB; reefs according to [8]	HUB	HUB
Predictors used for modelling	-	-	Only “Seagrass meadows and other marine macrophyte populations” and “Baltic aphotic muddy sediment dominated by ocean quahog (*Arctica islandica*)” were modelled in this study; the former is equivalent in their spatial extent to HUB class “Baltic photic mixed substrate dominated by perennial non-filamentous corticated red algae” and “Baltic a-/photic mixed substrate/coarse sediment dominated by foliose red algae” (*Zostera* spp. and *Fucus* spp. were not modelled in this study) and only indicated outside the reef areas; for predictors, see HUB entries	See detail areas	**Endobenthos:** sediment distribution map (50 × 50 m), water depth (50 × 50 m), temperature, salinity, current velocity (in directions north/south, east/west, without directional information), bottom shear stress, oxygen concentration, number of hypoxic days, DOC, ammonium, nitrate, phosphate (600 × 600 m) **Epibenthos:** boulder distribution map, water depth, photic zonation, slope gradient (50 × 50 m), temperature, salinity, current velocity (in directions north/south, east/west, without directional information), bottom shear stress, photosynthetically active radiation (PAR), oxygen concentration, number of hypoxic days, DOC, ammonium, nitrate, phosphate (600 × 600 m)	See detail areas; Tauber [32] was used instead of the sediment distribution map for endobenthos modelling, and reef coverage (as reported to HOLAS III) was used instead of boulder distribution map for epibenthos modelling
“Seagrass meadows and other marine macrophyte populations” (paragraph §30 Federal Nature Conservation Act)	-	-	Seagrass mapped in the “Plantagenet Ground”; other macrophytes modelled in this study	*Zostera* spp. modelled in Schleswig-Holstein [40] and Mecklenburg-Western Pomerania [41]; *Fucus* spp. modelled in Schleswig-Holstein [40]; other macrophytes modelled in this study	See OHT	See OHT

**Table 3 biology-13-00006-t003:** Translation of sediment types classified according to Tauber [32], following Folk [31] and Figge [36], into sediment classification according to EUNIS, on which the BHTs are based.

Sediment Type Classified according to Tauber (2012)	Sediment Type Reclassified according to EUNIS
gravel, very coarse sand	coarse sediment
fine sand—coarse sand	sand
very fine mud—very fine sand	mud
clay, peat, lag sediment/till	mixed sediment (hard substrate)

**Table 4 biology-13-00006-t004:** Number and sampling instruments of internal and external (in brackets) data that were mapped and acquired. Further data were obtained from the Federal Maritime and Hydrographic Agency (BSH), the State Office for the Environment, Nature Conservation and Geology of Mecklenburg-Western Pomerania (LUNG), the Schleswig-Holstein State Office for the Environment (LFU), the State Office for Agriculture and the Environment of Central Mecklenburg (StALU MM) and Western Mecklenburg (StALU WM), the Waterways and Shipping Office Stralsund (WSA Stralsund), the Christian-Albrechts-University Kiel (CAU Kiel), the Institute for Applied Ecosystem Research Ltd. (IfAÖ) (Neu Broderstorf, Germany), and the GEOMAR—Helmholtz Centre for Ocean Research Kiel.

	Area	Number of Acquired Data Points	Sampling Instruments	References of Used Data
Detail areas	Outer Wismar Bay	85 (18) grab stations, 29 video stations, 6 video transects	Van Veen grab, SeaViewer, BaSIS	IOW, IfAÖ, LUNG, StALU WM, StALU MM
	Darss Sill	73 (106) grab stations, 26 video stations, 4 video transects	Van Veen grab, SeaViewer, BaSIS	IOW, IfAÖ, LUNG, StALU WM, StALU MM
	Plantagenet Ground	49 (67) grab stations, 27 video stations, 4 video transects	Van Veen grab, SeaViewer, BaSIS	IOW, IfAÖ, LUNG, StALU WM, StALU MM
	Kadet Trench	103 (17) grab stations, 37 video stations, 8 video transects, 36 photo stations	Van Veen grab, SeaViewer, BaSIS, BfN drop camera	IOW, CAU Kiel, BSH
	Fehmarn Belt	339 grab stations, 134 video stations, 11 video transects	Van Veen grab, BaSIS	IOW, CAU Kiel, BSH
	German Baltic Sea	1637 (1991) grab stations, 403 video stations, 47 video transects, 59 photo stations, (45) diver stations, 9 (82) diver photo stations	Van Veen grab, SeaViewer, BaSIS, BfN drop camera, diver scratch samples and photos	IOW, BfN, BSH, LFU, LUNG, StALU WM, StALU MM, WSA Stralsund, CAU Kiel, IfAÖ, Geomar

**Table 5 biology-13-00006-t005:** Areas and their proportions of individual broad habitat types (BHT) in the German Baltic Sea.

BHT	Area (km^2^)	Area (%)
Infralittoral rock and biogenic reef	1.0	0.007
Infralittoral mixed sediment (hard substrate)	1785.3	11.6
Circalittoral mixed sediment (hard substrate)	488.3	3.2
Infralittoral coarse sediment	35.5	0.2
Circalittoral coarse sediment	16.2	0.1
Infralittoral sand	4600.0	29.8
Circalittoral sand	3010.4	19.5
Infralittoral mud	1393.8	9.0
Circalittoral mud	4115.1	26.6

**Table 6 biology-13-00006-t006:** Areas and their proportions of other individual habitat types (OHT) in the German Baltic Sea.

OHT	Area (km^2^)	Area within the German Baltic Sea (%)
Reefs (habitat type 1170)	2183.5	14.1
Sandbanks (habitat type 1110)	875.6	5.7
Seagrass meadows and other marine macrophyte populations	321.4	2.1
Species-rich areas of gravel, coarse-sand, and shell-gravel areas	5.9	0.04
Baltic aphotic muddy sediment dominated by ocean quahog (*Arctica islandica*)	1417.6	9.2
Non-OHT	10,641.6	68.9

**Table 7 biology-13-00006-t007:** Mapped HELCOM HUB biotopes and their respective areas with colour indication, as shown in the HUB map (Figure 6). The question marks (?) represent unclassifiable sediment areas. Unclassifiable sediment areas, presumably representing lag sediment areas and/or mussel beds with or without glacial till, were labelled as “AA.I1E1?” (without boulders) or as “AA.M1E1?” (with boulders). The codes marked with an asterisk (*) were introduced in this study and do not yet exist in the HUB classification.

Colour Coding HUB Map	HUB Code	HUB Biotope	Area (km^2^)
	AA.?	Baltic photic benthos	0.4
	AB.?	Baltic aphotic benthos	0.5
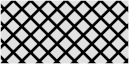	AA.?1E1	Baltic photic unknown substrate dominated by *Mytilidae*	0.5
	AA.?3	Baltic photic unknown substrate characterised by macroscopic infaunal biotic structures	2.1
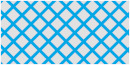	AA.?3L3	Baltic photic unknown substrate dominated by ocean quahog (*Arctica islandica*)	0.003
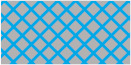	AB.?3L3	Baltic aphotic unknown substrate dominated by ocean quahog (*Arctica islandica*)	0.1
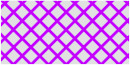	AA.?3L4	Baltic photic unknown substrate dominated by sand gaper (*Mya arenaria*)	0.4
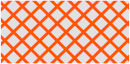	AA.?3L9	Baltic photic unknown substrate dominated by multiple infaunal bivalve species: *Cerastoderma* spp., *Mya arenaria*, *Astarte borealis*, *Arctica islandica*, *Macoma balthica*	2.4
	AA.M	Baltic photic mixed substrate	0.02
	AB.M	Baltic aphotic mixed substrate	0.04
	AA.M1	Baltic photic mixed substrate characterised by macroscopic epibenthic biotic structures	14.4
	AB.M1	Baltic aphotic mixed substrate characterised by macroscopic epibenthic biotic structures	15.3
	AA.M1C1	Baltic photic mixed substrate dominated by *Fucus* spp.	102.8
	AA.M1C2	Baltic photic mixed substrate dominated by perennial non-filamentous corticated red algae	16.3
	AA.M1C3	Baltic photic mixed substrate dominated by foliose red algae	840.8
	AB.M1C3 *	Baltic aphotic mixed substrate dominated by foliose red algae	0.9
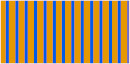	AA.M1C5	Baltic photic mixed substrate dominated by perennial filamentous algae	24.2
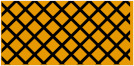	AA.M1E1	Baltic photic mixed substrate dominated by *Mytilidae*	540.6
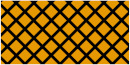	AA.M1E1?	Baltic photic mixed substrate dominated by *Mytilidae*?	0.2
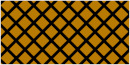	AB.M1E1	Baltic aphotic mixed substrate dominated by *Mytilidae*	302.9
	AA.M1G1	Baltic photic mixed substrate dominated by hydroids (*Hydrozoa*)	80.4
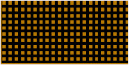	AB.M1G1	Baltic aphotic mixed substrate dominated by hydroids (*Hydrozoa*)	109.0
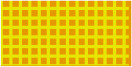	AA.M1H2	Baltic photic mixed substrate dominated by erect moss animals (*Flustra foliacea*)	0.02
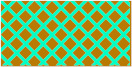	AB.M1I1	Baltic aphotic mixed substrate dominated by barnacles (*Balanidae*)	0.01
	AA.M1S1	Baltic photic mixed substrate dominated by filamentous annual algae	75.0
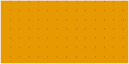	AA.M1V	Baltic photic mixed substrate characterised by mixed epibenthic macrocommunity	0.1
	AB.M1V	Baltic aphotic mixed substrate characterised by mixed epibenthic macrocommunity	10.9
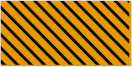	AA.M2T	Baltic photic mixed substrate characterised by sparse epibenthic macrocommunity	36.0
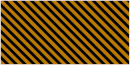	AB.M2T	Baltic aphotic mixed substrate characterised by sparse epibenthic macrocommunity	28.9
	AB.M4U	Baltic aphotic mixed substrate characterised by no macrocommunity	3.0
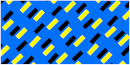	AA.G+AA.J1E1	Baltic photic peat bottoms + Baltic photic sand dominated by *Mytilidae*	1.0
	AA.I	Baltic photic coarse sediment	0.006
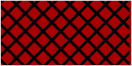	AA.I1E1	Baltic photic coarse sediment dominated by *Mytilidae*	11.4
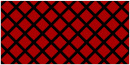	AA.I1E1?	Baltic photic coarse sediment dominated by *Mytilidae*?	0.7
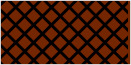	AB.I1E1	Baltic aphotic coarse sediment dominated by *Mytilidae*	9.0
	AA.I1C3	Baltic photic coarse sediment dominated by foliose red algae	0.1
	AA.I3	Baltic photic coarse sediment characterised by macroscopic infaunal biotic structures	10.5
	AB.I3	Baltic aphotic coarse sediment characterised by macroscopic infaunal biotic structures	3.9
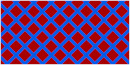	AA.I3L3 *	Baltic photic coarse sediment dominated by ocean quahog (*Arctica islandica*)	4.3
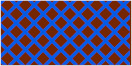	AB.I3L3 *	Baltic aphotic coarse sediment dominated by ocean quahog (*Arctica islandica*)	2.0
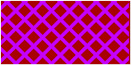	AA.I3L4 *	Baltic photic coarse sediment dominated by sand gaper (*Mya arenaria*)	0.2
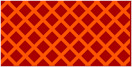	AA.I3L9 *	Baltic photic coarse sediment dominated by multiple infaunal bivalve species: *Cerastoderma* spp., *Mya arenaria*, *Astarte borealis*, *Arctica islandica*, *Macoma balthica*	4.1
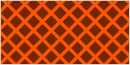	AB.I3L9 *	Baltic aphotic coarse sediment dominated by multiple infaunal bivalve species: *Cerastoderma* spp., *Mya arenaria*, *Astarte borealis*, *Arctica islandica*, *Macoma balthica*	1.7
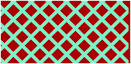	AA.I3L10	Baltic photic coarse sediment dominated by multiple infaunal bivalve species: *Macoma calcarea*, *Mya truncata*, *Astarte* spp., *Spisula* spp.	4.8
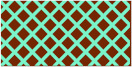	AB.I3L10	Baltic aphotic coarse sediment dominated by multiple infaunal bivalve species: *Macoma calcarea*, *Mya truncata*, *Astarte* spp., *Spisula* spp.	1.0
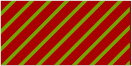	AA.I3L11	Baltic photic coarse sediment dominated by multiple infaunal polychaete species including *Ophelia* spp.	0.7
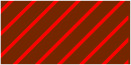	AB.I3M6 *	Baltic aphotic coarse sediment dominated by multiple infaunal polychaete species	0.01
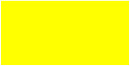	AA.J	Baltic photic sand	0.1
	AB.J	Baltic aphotic sand	0.005
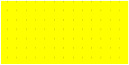	AA.J1B7	Baltic photic sand dominated by common eelgrass (*Zostera marina*)	223.1
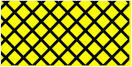	AA.J1E1	Baltic photic sand dominated by *Mytilidae*	141.3
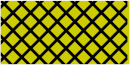	AB.J1E1	Baltic aphotic sand dominated by *Mytilidae*	196.2
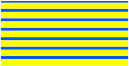	AA.J1S	Baltic photic sand characterised by annual algae	4.0
	AA.J3	Baltic photic sand characterised by macroscopic infaunal biotic structures	425.9
	AB.J3	Baltic aphotic sand characterised by macroscopic infaunal biotic structures	121.5
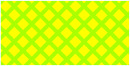	AA.J3L1	Baltic photic sand dominated by Baltic tellin (*Macoma balthica*)	8.2
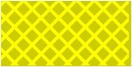	AB.J3L1	Baltic aphotic sand dominated by Baltic tellin (*Macoma balthica*)	60.7
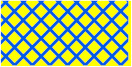	AA.J3L3	Baltic photic sand dominated by ocean quahog (*Arctica islandica*)	367.4
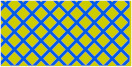	AB.J3L3	Baltic aphotic sand dominated by ocean quahog (*Arctica islandica*)	252.4
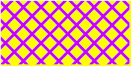	AA.J3L4	Baltic photic sand dominated by sand gaper (*Mya arenaria*)	15.7
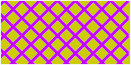	AB.J3L4	Baltic aphotic sand dominated by sand gaper (*Mya arenaria*)	0.1
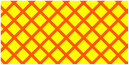	AA.J3L9	Baltic photic sand dominated by multiple infaunal bivalve species: *Cerastoderma* spp., *Mya arenaria*, *Astarte borealis*, *Arctica islandica*, *Macoma balthica*	2338.7
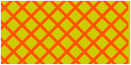	AB.J3L9	Baltic aphotic sand dominated by multiple infaunal bivalve species: *Cerastoderma* spp., *Mya arenaria*, *Astarte borealis*, *Arctica islandica*, *Macoma balthica*	2381.1
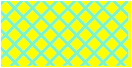	AA.J3L10	Baltic photic sand dominated by multiple infaunal bivalve species: *Macoma calcarea*, *Mya truncata*, *Astarte* spp., *Spisula* spp.	1.1
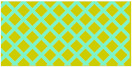	AB.J3L10	Baltic aphotic sand dominated by multiple infaunal bivalve species: *Macoma calcarea*, *Mya truncata*, *Astarte* spp., *Spisula* spp.	1.2
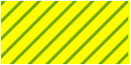	AA.J3L11	Baltic photic sand dominated by multiple infaunal polychaete species including *Ophelia* spp.	5.7
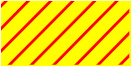	AA.J3M6*	Baltic photic sand dominated by multiple infaunal polychaete species	0.005
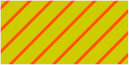	AB.J3M6*	Baltic aphotic sand dominated by multiple infaunal polychaete species	0.3
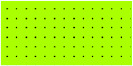	AA.H1B7	Baltic photic muddy sediment dominated by common eelgrass (*Zostera marina*)	69.0
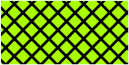	AA.H1E1	Baltic photic muddy sediment dominated by *Mytilidae*	46.6
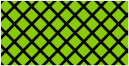	AB.H1E1	Baltic aphotic muddy sediment dominated by *Mytilidae*	15.0
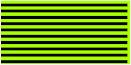	AA.H1S	Baltic photic muddy sediment characterised by annual algae	1.0
	AA.H3	Baltic photic muddy sediment characterised by macroscopic infaunal biotic structures	65.2
	AB.H3	Baltic aphotic muddy sediment characterised by macroscopic infaunal biotic structures	546.4
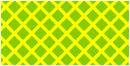	AB.H3L1	Baltic aphotic muddy sediment dominated by Baltic tellin (*Macoma balthica*)	1131.6
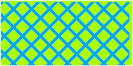	AA.H3L3	Baltic photic muddy sediment dominated by ocean quahog (*Arctica islandica*)	249.9
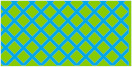	AB.H3L3	Baltic aphotic muddy sediment dominated by ocean quahog (*Arctica islandica*)	1435.5
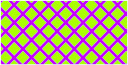	AA.H3L4 *	Baltic photic muddy sediment dominated by sand gaper (*Mya arenaria*)	0.003
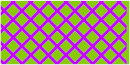	AB.H3L4 *	Baltic aphotic muddy sediment dominated by sand gaper (*Mya arenaria*)	0.005
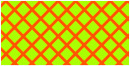	AA.H3L9 *	Baltic photic muddy sediment dominated by multiple infaunal bivalve species: *Cerastoderma* spp., *Mya arenaria*, *Astarte borealis*, *Arctica islandica*, *Macoma balthica*	307.9
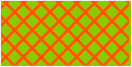	AB.H3L9 *	Baltic aphotic muddy sediment dominated by multiple infaunal bivalve species: *Cerastoderma* spp., *Mya arenaria*, *Astarte borealis*, *Arctica islandica*, *Macoma balthica*	991.0
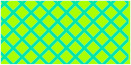	AA.H3L10 *	Baltic photic muddy sediment dominated by multiple infaunal bivalve species: *Macoma calcarea*, *Mya truncata*, *Astarte* spp., *Spisula* spp.	0.003
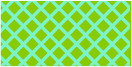	AB.H3L10 *	Baltic aphotic muddy sediment dominated by multiple infaunal bivalve species: *Macoma calcarea*, *Mya truncata*, *Astarte* spp., *Spisula* spp.	0.1
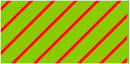	AB.H3M6	Baltic aphotic muddy sediment dominated by multiple infaunal polychaete species	3.2
		NA	8.0

**Table 8 biology-13-00006-t008:** Overall performance of the selected model in the respective detail areas and the German Baltic Sea; 95% CI: 95 % confidence interval of overall accuracy, AUC: area under (receiver operating characteristic) curve. Epibenthic communities did not need to be modelled in the “Plantagenet Ground”, as only mussels dominated the hard bottoms in this eastern area.

		Endobenthos					Epibenthos				
	Area	Overall Accuracy	95% CI	AUC	Kappa	Most Important Variables	Overall Accuracy	95% CI	AUC	Kappa	Most Important Variables
Detail areas	Outer Wismar Bay	0.393	0.215–0.594	0.758	0.035	current velocity (10th percentile)	0.98	0.893–1	0.975	0.96	DOC (10th percentile), O_2_ (10th percentile)
	Darss Sill	0.564	0.423–0.7	0.648	0.453	temperature (10th percentile)	1	0.936–1	1	1	DOC (mean)
	Plantagenet Ground	0.719	0.533–0.863	0.797	0.559	sediment	NA	NA	NA	NA	NA
	Kadet Trench	0.759	0.565–0.9	0.786	0.576	shear stress (mean), current velocity N/S (90 percentile)	0.657	0.556–0.748	0.716	0.485	depth
	Fehmarn Belt	0.763	NA	0.788	0.563	sediment, depth	0.926	NA	0.915	0.83	DOC (mean), depth
	German Baltic Sea	0.666	0.636–0.695	0.704	0.535	sediment	0.797	0.770–0.821	0.805	0.712	depth, salinity (mean)

## Data Availability

The BHT, OHT, and HUB maps are available as a map package (.mpk file) in the Appendix A. Soon, they will also be uploaded on EMODnet.

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
