# Peer review of "Habitats and Biotopes in the German Baltic Sea"

_biology, 2023, doi:10.3390/biology13010006_

Round 1

Reviewer 1 Report

Comments and Suggestions for Authors

The Baltic Sea is one of the most biologically studied sea basins in the world. The long history of study, combined with a relatively poor fauna, allows us to say that the communities and their ecology of this sea have been identified with sufficient completeness. However, as the article under review shows, there is a potential for greater insight into the issue of the Baltic Sea biological diversity. Although the work under review deals only with a part of the Baltic Sea (namely, the German one), the methodological (interdisciplinary) approach proposed by the authors, the results and their interpretation will be useful for researchers of marine biota in other areas of the Baltic, as well as other seas of the North Atlantic. Based on its quality, subject matter, and interest to the scientific community, the MS is certainly suitable for publication in the journal Biology. I believe that this manuscript can be accepted for publication almost in the form in which it was submitted to the editor, considering some minor comments (see my specific remarks below).

Lines 54-55. “Benthic habitats and their specific benthic communities (together considered as biotopes)”. A classical approach to the definition of a biotope is to define it as a sum of abiotic environmental parameters. Habitat + community constitute an ecosystem (or, possibly, an “association”, like in phytocoenology research), not a biotope. How the authors are free in choosing their terminology, I would recommend using a more common approach to the understanding of the biotope (This remark is not mandatory, of course).

Section 2.4 lacks an information on the total number of benthic samples analyzed. I believe that this publication is based on the analysis of a giant number of primary data collected at different times and using different methods. The paper would benefit if the total volume of material processed was presented in tabular form (or visualized in the form of graphs and charts). Table 4 fulfills this only partially.

Fig. 4. The legend includes “Infralittoral rock and biogenic reef” type, marked in blue. However, the map does not show this biotope type (no blue areas). I realize that this type is of very narrow distribution (only 0.007% of the total area). But the authors can indicate its situation by an arrow or by an asterisk (or a dot). Otherwise, why to include this type into the map legend?

Line 551. Mytilidae. The family name should not be italicized.

Author Response

Thank you for your constructive criticism. Please find attached my reply.

Reviewer 2 Report

Comments and Suggestions for Authors

Dear Editor and authors,

I have read your manuscript on habitats and biotopes in the German Baltic Sea with great interest. You have presented an impressive peace of work, and I completely agree with the emphasized need for the presented habitat and biotope maps. I also feel that the resulting maps are based on scientifically sound methods. However, the manuscript methods-section leaves me with quite some questions regarding the exact methodology applied, the rationale of several choices and their impact on the resulting maps. The results-section is already much clearer, but I believe that a better visualization of your results would improve the manuscript. For the discussion and conclusions, my overall feeling was that it was quite lengthy/wordy and that this should be shortened drastically. Please find attached a list with remarks / questions for the manuscript. In addition, the manuscript contains multiple instances where sentences are difficult to understand because sentences are too long, or English grammar is incorrect. I therefore really recommend a thorough read-through, preferably by a native speaker. 

Comments on the Quality of English Language

The manuscript contains multiple instances where sentences are difficult to understand because sentences are too long, or English grammar is incorrect. I therefore really recommend a thorough read-through, preferably by a native speaker. 

Author Response

Thank you for the very detailed and constructive criticism. Please find attached my reply.

Round 2

Reviewer 2 Report

Comments and Suggestions for Authors

Dear authors,

thank you for this revised manuscript. I am satisfied with the answers/adjustments you gave/performed with respect to my questions/remarks. The only comments I have left are editorial ones, as it seems that the layout of this manuscript has some (minor) issues (see underneath). I gladly recommend the editor to "accept in present form". All the best.

L 353: reference not found

L 417-431: text should not be in italics

L478 - 481: text should not be bold

L662 - 776: orientation should be 'portrait', not 'landscape'